# An Immunocompetent Environment Unravels the Proto-Oncogenic Role of miR-22

**DOI:** 10.3390/cancers14246255

**Published:** 2022-12-19

**Authors:** Maria Laura Centomo, Marianna Vitiello, Laura Poliseno, Pier Paolo Pandolfi

**Affiliations:** 1Department of Molecular Biotechnology and Health Sciences, University of Turin, 10126 Turin, Italy; 2William N. Pennington Cancer Institute, Renown Health, Nevada System of Higher Education, Reno, NV 89502, USA; 3Center for Genomic Medicine, Desert Research Institute, Reno, NV 89512, USA; 4Institute of Clinical Physiology, National Research Council, Via Moruzzi 1, 56124 Pisa, Italy; 5Oncogenomics Unit, Core Research Laboratory, ISPRO, Via Moruzzi 1, 56124 Pisa, Italy

**Keywords:** miR-22, oncomiR, animal models, genomic locus

## Abstract

**Simple Summary:**

Through their regulatory activity of gene expression, microRNAs have been heavily implicated in the genesis of cancer. They can act both as promoters (oncogenes) and suppressors of tumorigenesis. They can work within cells but also be secreted and uptaken by cells. While in certain instances, microRNAs appear to unequivocally act as oncogenes or tumor suppressor genes, in other instances, their activity appears to be tissue or context-dependent, in seemingly contradictory reports. MiR-22 has been extensively studied because of its powerful and diverse biological activities and has been attributed to both oncogenic and tumor-suppressive abilities. In this article, we critically analyzed this vast literature and reassessed the status of the miR-22 genetic locus in human cancer. We conclude that, when studied in immunocompetent model systems, miR-22 invariably acts as a tumor-promoting miRNA in view of its ability to impact the cancer immune microenvironment.

**Abstract:**

MiR-22 was first identified as a proto-oncogenic microRNA (miRNA) due to its ability to post-transcriptionally suppress the expression of the potent PTEN (Phosphatase And Tensin Homolog) tumor suppressor gene. miR-22 tumorigenic role in cancer was subsequently supported by its ability to positively trigger lipogenesis, anabolic metabolism, and epithelial-mesenchymal transition (EMT) towards the metastatic spread. However, during the following years, the picture was complicated by the identification of targets that support a tumor-suppressive role in certain tissues or cell types. Indeed, many papers have been published where in vitro cellular assays and in vivo immunodeficient or immunosuppressed xenograft models are used. However, here we show that all the studies performed *in vivo*, in immunocompetent transgenic and knock-out animal models, unanimously support a proto-oncogenic role for miR-22. Since miR-22 is actively secreted from and readily exchanged between normal and tumoral cells, a functional immune dimension at play could well represent the divider that allows reconciling these contradictory findings. In addition to a critical review of this vast literature, here we provide further proof of the oncogenic role of miR-22 through the analysis of its genomic locus *vis a vis* the genetic landscape of human cancer.

## 1. Introduction

MicroRNAs (miRNAs) are a class of small noncoding RNAs of ~22 nucleotides that emerged as critical post-transcriptional regulators of gene expression through translational repression or targeting mRNA for degradation [1]. Given their roles in regulating most protein-coding genes, miRNAs function in multiple biological processes, such as proliferation, differentiation, and apoptosis [2]. Thus, it is not surprising that miRNA dysregulation is associated with several human disorders [3], including cancer [4,5,6]. Compelling evidence has demonstrated that multiple mechanisms, including miRNA genomic alterations (i.e., amplifications, deletions, or mutations), transcriptional control changes, epigenetics alterations, and defects in miRNA biogenesis machinery (e.g., mutation or aberrant expression of Drosha, DGCR8, or Dicer) [7] drive abnormal expression and/or functioning of miRNAs in human cancer.

MiRNAs are involved in tumorigenesis as oncogenic factors (oncomiRs) or tumor suppressors by targeting and inhibiting specific cancer-related genes [8]. Generally, tumor-suppressor miRNAs are lost in cancer, whereas oncomiRs are overexpressed. In recent years, miRNA expression profiling and deep sequencing provided evidence that different tumors have specific signatures that can be used for tumor classification, diagnosis, and prognosis. It is important to note that the same miRNA can have different effects on various cell types and tissues because not all their targets are expressed at the same level in different tissues. Indeed, individual miRNAs may function as either oncogenes or tumor-suppressors in different tumors, but surprisingly even in the same tumor in specific biological contexts [9]. Among them, miR-22 has attracted considerable attention by playing various roles relating to multiple tumor types. Based on an extensive wealth of publications, miR-22 is dysregulated in various types of cancers and appears to act not only as an oncomiR but also as a tumor-suppressor, able to promote or inhibit tumor occurrence, progression and metastases through multiple processes [10,11,12,13,14,15]. While this might be in line with what is reported for other miRNAs and not entirely unexpected, it must be said that many of the publications reporting oncogenic or tumor suppressive roles for miR-22 have been solely based on in vitro studies performed on a limited number of cell lines. In a few cases, in vivo analyses have been entertained, and in even fewer cases, genetically modified and immunocompetent animal models (e.g., knock-out and/or transgenic mouse models) have been utilized. With this in mind, we will try to discern and review this complex landscape of publications, emphasizing the instances in which a robust analysis of what is observed in human cancer specimens and human cell lines is accompanied by rigorous validation in animal models.

Unlike most other miRNAs, which are grouped in multiple-member families, miR-22 belongs to a single-member miRNA family, and it‘s evolutionally conserved from flies to mammals. Mir-22 is encoded by exon 2 of the long non-coding miR-22 host gene (MIR-22HG), which consists of four exons and is located on Chromosome 17p13, a frequently deleted, hypermethylated or loss of heterozygosity (LOH)-associated region [16,17,18] as we will discuss below (Figure 1). Mir-22 is ubiquitously expressed, and many studies have revealed that this miRNA can regulate various cellular processes, getting involved in many different disorders other than cancer, such as cardiovascular, neurodegenerative, inflammatory, autoimmune, and metabolic diseases [19,20,21,22,23,24]. Mir-22′s role in these conditions is well established and agreed upon within the scientific community (e.g., miR-22 is a master regulator of lipid metabolism and anabolic rewiring [23,25,26]). The outlook is seemingly much more complex in cancer. As mentioned above, the current literature suggests that miR-22 may play an important role in cancer both as a tumor suppressor and oncomiR in different contexts, as also evidenced by the large number of cancer-relevant direct targets of miR-22 known to act as bona fide oncogenes and tumor suppressors genes (Table 1 lists currently validated, cancer-relevant targets of miR-22). This potentially opposite role (promotion/suppression) of miR-22 on the biological behavior of cancer is quite evident in the EMT process in which miR-22 may either work to trigger or inhibit EMT. Mir-22 triggers EMT by targeting Zeb1/2, Bmi1, and E-cadherin and inhibits EMT by targeting MMP14 and Snail [11,27,28]. Therefore, the effect of the miR-22 function seems to be context-dependent and drives different outcomes according to cell type and gene expression pattern. However, a completely different scenario emerges when paying attention to the fact that the role of miR-22 is coherently proto-oncogenic when studied in vivo in immune-competent model systems. On the contrary, its role is utterly confusing and even contradictory when in vitro cell culture analysis or in vivo immune deficient xenograft model systems are employed.

For instance, reviewing the literature from 2010 to date, a quite consistent role for miR-22 as a tumor suppressor has been reported in several tumors (e.g., colon cancer, gastric cancer, lung cancer, osteosarcoma…), based on several in vitro and in vivo studies (Table 2). It’s worth noticing that all these in vivo reports are based on xenograft mouse models. However, the strong influence of the tumor microenvironment on cancer initiation, progression, and metastasis makes the immunocompetent genetically engineered mouse models (GEMMs) a much greater reliance. Indeed, when true genetic mouse models have been employed, a clear role of miR-22 as an oncogene (e.g., in prostate cancer) definitely emerged. The same is true in tumors such as breast cancer and leukemia, where at least one transgenic mouse model-based study has been conducted (Table 2). Strong evidence derived from these GEMM-based studies demonstrates that miR-22 is a crucial driver in promoting cancer development and aggressiveness, by targeting and inhibiting tumor-suppressor genes, such as PTEN and TET (Ten eleven translocation) methylcytosine dioxygenase family members, thus upregulating cancer-associated pathways [11,29,30] (Figure 2). Conversely, in the same tumor context, PDX-based studies show that miR-22 is capable of acting as a tumor-suppressor by directly or indirectly targeting downstream proto-oncogene transcription factors, thus suppressing cell growth and invasion [31,32,33].

A first work finally came up establishing, in a rather indisputable way, the oncomiR function of miR-22 in promoting cutaneous squamous cell carcinoma (cSCC) initiation, progression, and metastasis [34]. Combining a miR-22 knock-out and a chemically (DMBA/TPA)-induced cSCC mouse model, [34] reports that loss of miR-22 hinders the tumorigenesis also by targeting tumor suppressors FOSB and PAD2, thus maintaining Wnt/β-catenin signaling and cancer stem cells (CSCs) function compared to WT mice (Figure 2a). Therefore, miR-22 can function as an oncogene because of its antagonistic effect on tumor-suppressive pathways.

Epigenetic alterations have profound effects on tumor development, progression, and resistance to therapy. In keeping with this notion, miR-22 was found to be an epigenetic modifier controlling cancer growth and proliferation through the modulation of histone acetylation, DNA methylation, and gene repair. Moreover, in this setting, if we take a look at the literature published so far, miR-22 seems to have a dual role, depending on the target and the tumor. However, as aforementioned, these seemingly conflicting roles of miR-22 may also be due to the extensive use of cancer cell lines and in vitro analyses rather than the use and analysis of genetically modified cancer models *in vivo*. Indeed, in leukemia and breast cancer, miR-22 exerts its oncomiR function promoting stem cells’ self-renewal, transformation, and metastasis, at least in part through direct targeting of TET family proteins [11,29], thus regulating 5-hmC (5-hydroxymethylcytosine) as strongly demonstrated by GEMM-based studies. In Hepatocellular Carcinoma (HCC) and other cancers (e.g., Ewing Sarcoma), the miR-22-mediated post-transcriptional silencing of histone deacetylases HDAC4 [35] or KDM3A (lysine (k)-specific demethylase 3A) [36], with consequent inhibition of tumorigenesis, has been evidenced, once again, by in vitro and in vivo xenograft models, using immunocompromised mice.

Here we critically review the literature focusing on in vivo studies while highlighting what is still missing about the controversial role of miR-22. Particularly, we focus on three different tumors as paradigmatic examples (breast cancer, prostate cancer, and leukemia), where the role of miR-22 is strongly supported by genetic mouse model-based studies. Finally, we also analyze the genomics landscape of miR-22 in human cancer.

## 2. Breast Cancer

Several reports have been published about the role of miR-22 in breast cancer (about 60). Some of them describe miR-22 as an oncogene [29,37,38,39,40,41], and some others as an oncosuppressor [31,42,43,44,45,46,47,48,49,50,51]. There is then the extreme case of [41], where miR-22 has been shown to act both as a tumor suppressor and oncogene in the same cell line. This work has been conducted on fulvestrant-resistant MCF-7/182R-6 breast cancer cell line. These cells are characterized by a moderate upregulation of miR-22 compared to fulvestrant-sensitive cells. In this context, both the inhibition and the overexpression of miR-22 suppress proliferation, induce apoptosis, cause S-phase arrest, and enhance the expression of p21. However, when overexpressed, miR-22 leads to p21 upregulation by targeting the transcriptional repressor FOXP1. Conversely, when inhibited, miR-22 leads to p21 upregulation through p53.

Assuming that in vivo models, where different cells, organs, and systems can interact all together, are the most accurate setting to establish the function of a gene of interest, we focus on the two main examples of miR-22-based mouse models: [29,39] both define miR-22 as an oncogene in breast cancer.

According to [29], miR-22 triggers EMT, enhances cell proliferation, cell motility, invasiveness, and angiogenesis, and promotes metastasis. In addition to in vitro and in vivo xenograft models, conditional transgenic mouse models have been used: (1) miR-22F/+;MMTV-Cre, (2) MMTV-PyVT;miR-22F/+;MMTV-Cre and (3) MMTV-neu;miR-22F/+;MMTV-Cre mice. miR-22F/+;MMTV-Cre model: mice that harbor, within their Collagen A1 locus, the CAGGS promoter and the miR-22 genomic sequence separated by a LoxP flanked transcriptional STOP cassette (miR-22F/+) have been crossed with mice that express the Cre recombinase under the control of the mouse mammary tumor virus promoter (MMTV-Cre). In this way, it has been possible to overexpress miR-22 selectively in the mammary glands. MMTV-PyVT;miR-22F/+;MMTV-Cre model: MMTVPyVT transgenic mice, which develop multifocal mammary tumors that spontaneously metastasize to the lung [52], have been crossed with miR-22F/+;MMTV-Cre mice. Thanks to this mouse model, the authors found that the penetrance of metastatic cancer is increased in MMTV-PyVT;miR-22F/+, MMTV-Cre mice when compared to controls. MMTV-neu;miR-22F/+;MMTV-Cre model: MMTV-neu mice express inactivated neu (c-ErbB2) and show less aggressive mammary tumors and lung metastases than MMTV-PyVT animals. Thanks to this mouse model, the authors found that the development of primary mammary gland lesions, as well as the incidence of lung metastases, is significantly increased in MMTV-neu; miR-22F/+;MMTV-Cre mice when compared to controls. They also showed EMT-related breast tumor phenotypes. In summary, the aforementioned animal models indicate that miR-22 enhances mammary gland side-branching, expands the stem cell compartment, and promotes tumor development, as well as aggressive metastatic disease.

miR-22 exerts its metastatic potential by silencing anti-metastatic miR-200 through direct targeting of the TET family of methylcytosine dioxygenases, thereby inhibiting the demethylation of miR-200 promoter. miR-200 family members are known as tumor suppressive miRNAs that regulate the EMT and control stemness by directly targeting transcriptional repressor Zeb1/2 and polycomb repressor complex proteins, such as Bmi1 and Suz12 (Figure 2b).

The other in vivo demonstration of the oncogenic role of miR-22 is reported in a recent work about tamoxifen resistance [39]. In addition to the direct study of breast cancer cells, the tumor microenvironment is considered as well. Particularly, it has been found that CD63+ cancer-associated fibroblasts (CAFs) secrete exosomes rich in miR-22. These vesicles are then uptaken by breast cancer cells, which in fact, express higher levels of miR-22 when cocultured with CD63+ CAF. Once inside cancer cells, miR-22 targets ER*α* and PTEN, conferring tamoxifen resistance. MMTV-PyVT;miR-22−/− mice are, in fact, more sensitive to tamoxifen than MMTV-PyVT;miR-22+/+ mice (FVB/N-Tg(MMTV-PyMT)634Mul/J transgenic mice [52,53] were crossed with miR-22 knockout (KO) mice (model from Nanjing University; C57BL/6N background)). This transgenic mouse model is the same used by [29], but miR-22 is deleted in homozygosity (−/−)and also total body. Accordingly, when a miR-22 sponge is used in CD63+ CAFs, a decrease in miR-22 expression in exosomes is observed, and consequently, its suppressive effect on ER*α* expression and its ability to induce tamoxifen resistance is compromised. For this reason, the authors propose to use cRGD-miR-22-sponge nanoparticles as a targeted delivery system for breast cancer cells. In such nanoparticles, cyclic RGD (cRGD), a “tumor-homing” cyclic peptide, binds directly to *αβ* integrin and carries a miR-22 sponge inside target cells. These nanoparticles are able to enhance the therapeutic effect of tamoxifen in MMTV-PyMT mice. Interestingly, Xiong et al. reported that inhibiting endogenous miR-22 in ER*α*-negative MDA-MB-231 cells could restore the expression of ER*α* [51]. These data confirm ER*α* as a strong target of oncogenic miR-22.

The results obtained in mouse models are backed up by data on breast cancer patients. Very few are the datasets where miR-22 has been reported as downregulated [49,54], with the majority of datasets reporting its upregulation [55]. Indeed, 13 out of 15 studies collected in the dbDEMC database show miR-22 as upregulated in breast cancer samples (Figure 3a,b). Moreover, it is also well established that high expression of miR-22 correlates with poor clinical outcomes in breast cancer patients (Figure 3c) [29,39,56].

Together, data collected from mouse models and patient samples strongly indicate that miR-22 plays an oncogenic role in breast cancer. Particularly, this is clearly demonstrated when the whole organism is considered using in vivo models.

## 3. Prostate Cancer

Prostate cancer is an example of a tumor where the oncogenic role of miR-22 appears indisputable. In the majority of the 15 papers published on the topic, miR-22 is, in fact, described as an oncogene [28,30,57,58,59,60,61,62] or even proposed as a biomarker [58,61].

One of the first papers that well describes the role of miR-22 in prostate cancer is [30]. Here, miR-22 has been shown to target the 3′UTR of PTEN (Figure 2e), as confirmed by other researchers later on [62]. miR-22 cooperates with the proto-oncogene cMYC in MEF transformation. Moreover, when prostate cancer cells overexpressing miR-22 are injected into the flank of nude mice, there is a proliferative advantage in tumor growth.

In agreement with Pandolfi’s group, Levenson’s group [28] identified miR-22 as an oncomiR belonging to the class of Epi-miRs that regulates EMT. Particularly, by Chip-seq studies, metastasis-associated protein 1 (MTA1), a chromatin remodeler, has been found to directly regulate miR-22. miR-22, in turn, targets the 3′UTR of E-Cadherin, promoting prostate cancer cell invasiveness and migration. Indeed, prostate-specific overexpression of MTA1 in the mouse (Pb-CRE+; R26 MTA1 mice) leads to an increase in miR-22 expression and a decrease in E-Cadherin expression (Figure 2c). Moreover, [28] has observed a positive correlation between miR-22 levels and MTA1 RNA/protein levels and a negative correlation between miR-22 levels and E-Cadherin RNA/protein levels in a cohort of prostate adenocarcinoma biopsies. The oncogenic role of miR-22 has also been described indirectly: when prostate neoplasia is attenuated by adding grape powder (proposed as a chemopreventive strategy for prostate cancer progression) to the diet of prostate-specific Pten+/− mice (C57BL/6J;Ptenf/f female mice crossed with B6.Cg;Pb-Cre4 male mice), the circulating levels of oncogenic miR-22 are reduced [59]. Similarly, lower miR-22 expression levels have been observed in mice fed with a pterostilbene-supplemented diet, which exhibit more favorable histopathology, with decreased severity and number of PIN foci, accompanied by reduced proliferation, angiogenesis, and inflammation [57].

Importantly, miR-22 has been found overexpressed both in primary and metastatic prostate carcinomas when compared with normal epithelium (frozen prostate samples were dissected via laser capture microdissection into benign glandular epithelial versus tumor [62]). Analogously, it has been found overexpressed in a prostate tumor microarray containing 184 cases of human tumor and matched non-tumor tissues [30].

Very few works describe miR-22 as a tumor suppressor miRNA in prostate cancer: [63,64]. Although this last in vitro study appears to be in contradiction with what is described above, this is not entirely the case. Ref. [64] have found that Androgen Receptor (AR) regulates miR-22 transcription (AR binds MIR22HG at three different sites, all close to the transcription start site), and in turn, miR-22 negatively regulates LAMC1 and Mcl-1 (both oncogenes). However, when they screen a panel of prostate cancer cell lines (DU145, PC3, PC3AR, VCaP, DUCaP, LNCaP, LAPC-4, CWR22RV1), they found that miR-22 is less abundant than in benign cell lines only in three of them (LNCaP, DUCaP, and CWR22RV1). Looking into these cell lines, they found that they are characterized by a high content of wild-type AR and robust AR transcriptional activity. Conversely, AR-negative cell lines, such as DU145 and PC3 (cell lines used by [30,64]), show levels of miR-22 that are enormously higher than in benign cell lines. Moreover, when they transfect miR-22, they observe a diminished cell migration in both AR-positive LNCaP and AR-negative PC3 cells but not an increase in cell apoptosis or decrease in cell viability [64]. On the contrary, PC3 cells show an increase in cell viability.

Overall, these results validate the oncogenic role of miR-22 in prostate cancer, at least in the context of AR-negative prostate cancer.

## 4. Leukemia

Hematopoiesis is a complex, multistep process finely orchestrated by several genetic and epigenetic mechanisms, whose deregulation is associated with leukemic transformation. Both myelodysplastic syndrome (MDS) and leukemias are frequently characterized by aberrant epigenetic modifications, including DNA methylation and histone modification, which in the case of MDS are typically associated with rapid progression to acute myeloid leukemia (AML) and poor prognosis [65,66]. Over time, different genes known to control hematopoiesis, such as TET1/2/3, IDH1/2, DNMT3, and EZH2, have been shown to affect the epigenetic landscape and are frequently mutated in patients with hematological malignancies [66].

It is also becoming evident that miRNAs acting in complex regulatory networks as regulators of normal hematopoietic differentiation may also contribute to aberrant hematopoiesis and leukemogenesis as either oncomiRs or tumor-suppressors [67,68,69]. MiRNAs not only act as post-transcriptional regulators of gene expression, but they can also be targets of the epigenetic machinery, as well as effectors of DNA and/or histone modifications [70,71,72], all functions likely involved in leukemogenesis. In hematological malignancies originating from different lineages, individual miRNAs can play distinct roles. However, evidence-based on in vivo mouse model studies has shown that some miRNAs, such as miR-22, may exert opposite effects in hematopoiesis functioning both as tumor suppressors and oncomiRs in leukemia development from the same hematopoietic lineage, such as in de novo AML and MDS [11,33]. Once again, however, these opposing results may be reconciled by the use of immune-competent versus immune-deficient experimental approaches, as we discuss below. In 2013, [11] demonstrated that miR-22 exerts proto-oncogenic activity in MDS and hematological malignancies by being strongly overexpressed and by negatively regulating TET2, a tumor-suppressor that is recurrently mutated or inactivated in a variety of human hematological tumors (i.e., MDS, myeloproliferative neoplasm (MPN), chronic myelomonocytic leukemia (CMMoL), and AML) [11,73,74,75,76]. On the contrary, a few years later, Jiang et al. reported that miR-22 is an essential tumor suppressor in most cases of de novo AML, where it is significantly downregulated [33], in agreement with what was reported by Shen et al. [77]. Therefore, how can miR-22 act both as an oncogene and a tumor suppressor in the same cell lineage? Once again, these opposing results may be reconciled by the use of immune-competent versus immune-deficient experimental approaches, as we discuss below.

In the first paper published, microarray profiling and comprehensive in situ hybridization analysis revealed that miR-22 is highly expressed in patients with MDS, and its aberrant expression correlates with a poor survival rate. These findings led the authors to further analyze the role of miR-22 in hematopoiesis and malignancy evolution. Through the generation of immunocompetent transgenic mice conditionally expressing miR-22 in the hematopoietic compartment (miR-22F/+;Mx1-Cre), they demonstrated that miR-22 enhances hematopoietic stem/progenitor cells (HSPCs) maintenance and self-renewal ability, triggering MDS-like syndrome and AML transformation over time. Mechanistically, they showed that miR-22 directly suppresses TET2 expression, affecting the global epigenetic landscape (e.g., 5-hydroxymethylcytosine (5-hmC) levels) and leading to the aberrant expression of putative TET2 target genes, such as AIM2 and SP140 (Figure 2d). Ectopic expression of TET2 antagonized the oncogenic function of miR-22, leading to a significant advantage in disease-free survival. Further, combined immunohistochemical and in-situ hybridization analysis revealed that most MDS patients and some AML patients with multilineage dysplasia (MLD) displayed reduced levels of TET2, which was directly anti-correlated with miR-22 expression level and associated with worse prognosis. This finding was also confirmed in 58.1% of primary AML patients. These data strongly suggest that miR-22 acts as an oncomiR to inactivate TET2 as an alternative mechanism in addition to mutation and deletion and that an aberrant miR-22/TET2 cross-talk is a common event in hematopoietic malignancies.

It is noteworthy that while [11] found an increased level of miR-22 in AML patients, both Jiang et al. and Shen et al. reported the opposite. Surprisingly, TET2 positively correlated with miR-22, whereas TET1 exhibited a negative correlation [33]. Therefore, in contrast to the reported miR-22 oncogenic function and overexpression in MDS, they showed that miR-22 has an antitumor effect in the pathogenesis of AML, and it is significantly downregulated in most de novo AML patients due to TET1/GFI1/EZH2/SIN3A-mediated epigenetic repression and/or DNA copy number loss. Specifically, using retrovirally transduced normal bone marrow transplantation (BMT) in a “lethally irradiated” C57BL/6 syngeneic mouse model, the authors found that forced expression of miR-22, including MSCV-PIG-miR-22_2 construct from [11], dramatically inhibits the leukemogenesis induced by MLL-AF9, by repressing both CREB and MYC signaling pathway. This was further confirmed by G7-NH2-nanoparticles carrying miR-22-based therapy. Conversely, miR-22 knock-out in primary and secondary BMT “lethally irradiated” recipient mice resulted in the opposite effect, promoting AML, thus emphasizing the critical role of miR-22 in both development and maintenance. As mentioned above, such repression of miR-22 expression in AML has been found to be related to deletions in the miR-22 gene locus on chromosome 17 and/or epigenetic regulation mechanism TET1 mediated [33,78]. Oncogenic fusion genes promote the expression of GFI1 and TET1, which in turn recruits polycomb cofactors (e.g., EZH2/SIN3A) at the miR-22 promoter, increasing H3K27me3 and decreasing RNA Pol II binding. This genetic/epigenetic inactivation of miR-22 results in the re-activation of its oncogenic targets, including CRT1, MYCBP, and FLT3, and thereby activating the CREB and MYC signaling pathway, accompanied by cell transformation and leukemogenesis.

Therefore, while the study of [11] supports strong evidence of a miR-22 oncogenic function in MDS and MDS-derived AML using transgenic mouse models, more recent studies support exactly the opposite in *de novo* AML. Among them, [33] efforts are the most convincing, given the use of both miR-22 gains of function and loss of function mouse models. Both studies display a vast number of controls that are very extensive and consistent. This Janus-face nature of miR-22 could be related to the complexity and the heterogeneity of the genetic/epigenetic landscapes of the MDS and MDS-derived AML, largely different from de novo AML and its subtypes. Indeed, a specific function of miRNA depends on the expression of its target, the accessibility of the 3‘UTR, and the functional relevance of that target in each cell line, which may be completely different in different hematopoietic stages and different leukemias. In this respect, the miR-22 expression has been found to vary profoundly in AML patients. Additionally, it‘s worth considering that gene expression profiles greatly differ between various cell types. Therefore, the separation of one cell type is essential for accurate gene expression profiling. As MDS and AML originate from HSPCs, the analysis of CD34+ HSPCs fraction is of primary interest, as performed by [11]. Likewise, mononuclear cells (MNCs) isolated from the AML patients’ bone marrow (BM), or peripheral blood (PB) cells were used in the [33] work. Moreover, the studies from [11,33] provide insight into the vast genetic/epigenetic differences that exist between AML and MDS. Once again, the immune dimension may be a decisive differentiating factor: in the [11] study, mice were immunocompetent, while in the [33] study, they were immunodeficient.

## 5. Alterations of miR-22 Gene

MiRNA genes’ location in the genome is nonrandom [16]. Indeed, a significant number of miRNAs (about 19%) are located in or very close (< 3Mb) to fragile sites or, alternatively, to Human Papilloma Virus Integration sites [16]. Examples are chromosome 17 (where miR-22 is located) and 19, which contain more miRNAs than expected based on their size, while chromosome 4 contains less [16].

Chromosome alterations such as deletions, amplifications, and/or mutations are frequent events in cancer cells. Particularly, amplifications of loci-containing oncogenes and deletions of loci-containing tumor suppressors have been frequently founding [79]. Analogously, the overexpression of oncogenes and the downregulation of tumor suppressor genes are frequently observed.

MicroRNAs are no exception to this general rule, and in this paragraph, we have undertaken a systematic analysis of miR-22 genome locus/gene expression in order to gather further indications in support of its oncogenic role.

MiR-22 (Chromosome 17: 1,713,903–1,713,987) is located in the 17p13.3 locus together with its host gene, the lncRNA HG-miR-22 (Chromosome 17: 1,711,447–1,717,174). Interestingly, telomeric to miR-22 gene there are known oncogenes such as CRK (Chromosome 17: 1,420,689–1,463,162) [80] and PITPNA-AS1 (Chromosome 17: 1,516,877–1,518,101) [81,82,83]. Conversely, centromeric to miR-22, there are several known tumor suppressor genes, such as miR-212 (Chromosome 17: 2,050,271–2,050,380), miR-132 (Chromosome 17: 2,049,908–2,050,008) and HIC1 (Chromosome 17: 2,054,154–2,063,241), which are all in the same cluster, and are very close to each other, as well as to TP53 (Chromosome 17: 7,661,779–7,687,538) (Figure 1). Particularly, miR-212 and miR-132 are known tumor suppressor miRNAs in various cancers, including lung cancer [84], hepatocellular carcinoma [85], prostate cancer [86], and glioblastoma [87,88].

miR-22 is in a genomic region frequently altered, and if we compare miR-22 amplification/deletion ratio with that of miR-132 or miR-212, it is possible to observe some interesting differences, in spite of the fact that they are located only 336 kbp apart. Table 3 describes the number of amplifications and deletions reported in the cBioportal database (cbioportal.org) for the three microRNAs in all tumors and, specifically, in breast cancer, hepatocellular carcinoma (HCC), and in prostate cancer. In all cases, the ratio between the number of amplifications and deletions observed for miR-22 is higher than that observed for the other two miRNAs (see also Figure 3d). This suggests that the miR-22 gene is more predisposed to be amplified than deleted when compared to the other two miRNAs (that are known tumor suppressor genes), and this is clearer in cancer types, such as prostate cancer, where the oncogenic role of miR-22 is well established.

A similar result is obtained when the ClinVar database of NCBI, which contains information relative to all types of tumors, is interrogated: 33 deletions and 38 amplifications are reported for miR-22 genomic locus vs. 33 deletions and 28 amplifications for miR-132 and miR-212 genomic locus.

Both in cBioportal and in ClinVar, the number of cases is very low. However, the number of amplifications and deletions reported for the three miRNAs is more or less the same, and a higher number of amplifications together with a lower number of deletions for miR-22, when compared with miR-132 and miR-212, is consistently reported. These data support the “less” oncosuppressive role of miR-22 compared to the other two miRNAs.

Moreover, looking at miR-22, miR-132, miR-212, miR-22HG, and HIC1 genes on the cBioportal database, all five genes show co-occurrence in genetic alterations. Indeed, in the majority of cases, alterations are absent (*Neither* column) or present (*Both* columns) in all the analyzed genes (Table 4). However, comparing two genes at the time when a genomic alteration is present in at least one of them (*A Not B*, *B Not A*, and *Both* columns), miR-132 and miR-212 (3rd row) always show co-occurrent alterations. More precisely, among the 213 cases where at least one of the two miRNAs is altered, there are only 2 cases in which miR-132 is the only one to be altered, while in the remaining 211 cases, they are co-altered. A similar trend is observed between HIC1 and miR-132, as well as between HIC1 and miR-212 (8th and 9th rows). As for miR-22, a co-occurrence of alterations with miR-132, miR-212, and HIC1 is observed as well (1st, 2nd, and 7th row, respectively), but the cases where only one of them is altered are more than those counted for miR-212 and miR-132. The results reported in the 1st and 2nd rows are statistically different from those reported in the 3^rd^ row (1st vs. 2nd row: fisher exact test = 0.921, *p*-value > 0.05; 2nd vs. 3rd row: fisher exact test < 0.00001; *p*-value < 0.05; 1st vs. 3rd row: fisher exact test ≤ 0.00001, *p*-value < 0.05).

In Figure 3e, we report, as an example, the number of amplifications and deletions of miR-22, miR-132, and miR-212 in breast cancer patients, according to the cBioportal database. The bar graph shows how gene deletions and amplifications are distributed across the three miRNA genes. Particularly, it highlights the predominant number of patients where miR-132 and miR-212 deletions co-occur with no alteration of miR-22 (black square).

Altogether, these results point toward a different alteration pattern for miR-22 when compared to miR-132 and miR-212. Such a difference is, in turn, consistent with the fact that miR-132, miR-212, and HIC1 genes are close to each other, while the miR-22HG gene is farther away toward the telomeric end of chromosome 17.

Building upon this, we can hypothesize that, as shown in Figure 1, miR-22 is located at the verge of two regions: one more telomeric, which is oncogenic, and one more centromeric, which is tumor suppressive. Moreover, we should keep in mind that the chromosome 17p13 region (where miR-22 is located) contains genes such as TP53 (Chromosome 17: 7,661,779–7,687,538), which is a powerful tumor suppressor. Therefore, there are several selective forces that play all together to break homeostasis towards tumorigenesis.

To verify our hypothesis about the two different regions on chromosome 17p13.3, we have extended the region of study, and we have compared the genomic alterations (deletions and amplifications) of the HIC1 gene (which, as mentioned above, is located centromeric to miR-132 and is a known tumor-suppressor) with those of two known oncogenes located telomeric to miR-22: *CRK* and *PITPNA-AS1* (Figure 1). When looking at the numbers of deletions and amplifications according to the cBioportal database, it is possible to observe that the p13.3 portion of chromosome 17 (where these three genes are all located) is more frequently deleted (112 cases) than amplified (73) (Figure 3f, black arrows). This agrees with the fact that chromosome 17p13.3 is rich in tumor-suppressor genes, as we mentioned above. In particular, the three genes are generally amplified (Figure 3f, left, black arrow) or deleted (Figure 3f, right, black arrow) as one block. However, considering genetic alteration events that involve two genes at a time, we can observe that *CRK* and *PITPNA-AS1* are more frequently co-altered than altered together with HIC1. Moreover, when they are co-altered, they are predominantly amplified (Figure 3f, left, red arrow). This point supports our hypothesis of two different regions of chromosome17p13.3, with that encompassing *CRK*, *PITPNA-AS1,* and miR-22 being oncogenic.

Using the GenomAD browser (“https://gnomad.broadinstitute.org/ (accessed on 15 November 2022)”), we have also analyzed the exon sequences of miR-22 vs. miR-132 and miR-212 oncosuppressive miRNAs for the presence of annotated base mutations. As shown in blue in Figure 3g, miR-22 is the miRNA with fewer reported base pair variations in its mature sequence (only 2 in the miR-22-5p sequence, see Appendix A-3rd and 4th rows). Notably, no variation has been found in the miR-22-3p sequence. On the contrary, miR-212, as well as miR-132, are characterized by a high number of base pair variations in their mature sequences (17 for miR-212 and 7 for miR-132). Analogous results were obtained by analyzing the three pre-miRNA sequences: the number of variations identified in pre-miR-132 and pre-miR-212 is much higher than that of variations identified in pre-miR-22 (compare Appendix A). Moreover, allele frequencies of each variant reported for miR-132 and miR-212 are higher (Appendix A; column j). This, in turn, means that miR-132 and miR-212 genome sequences are more prone to variations than that miR-22. Unfortunately, we cannot establish the effects of each variation on the transcription, maturation, and functionality of the miRNAs. Nonetheless, we can assume that the higher variation rate is in line with the tumor suppressive role of miR-212 and miR-132.

Finally, starting from the assumption that in cancer, an overexpressed gene likely acts as an oncogene, while a downregulated gene likely acts as a tumor suppressor, we have analyzed miR-22 expression in different cancers thanks to dbDEMC 3.0 (Database of Differentially Expressed miRNAs in Human Cancers; https://www.biosino.org/dbDEMC/index; (accessed on 15 November 2022)” [89]). This repository collects expression levels of miRNAs in cancer, detected by high-throughput methods such as microarray or miRNA-seq, and reported in public repositories, including Gene Expression Omnibus (GEO), Sequence Read Archive (SRA), ArrayExpress, and The Cancer Genome Atlas (TCGA). According to dbDEMC 3.0, miR-22 is upregulated in the majority of studies in breast cancer, where it acts as an oncogene, while it is downregulated in hepatocellular carcinoma and lung cancer, where it has been described to have a tumor suppressor role. Furthermore, dbDEMC 3.0 indicates that in breast cancer, miR-22 is upregulated, while miR-132 and miR-212 are downregulated, as expected (Figure 3a). In addition to this data, a study performed on 36 pairs of tumor and matched non-tumor specimens (marginal nontumor counterparts) from human patients with breast invasive ductal carcinoma confirms the overexpression of miR-22 and the downregulation of miR-212 and miR-132 [55]. More specifically, when tumors are divided according to their stage, miR-22 expression level is higher in stage III breast cancer samples compared to stage I-II. Conversely, the higher the grade of the tumor, the stronger the downregulation of miR-212 and miR-132 expression. Unfortunately, the authors of this work have not well described how they perform the qRT-PCR, particularly they have not reported the sequence of the primers used nor explained if they amplify pre-miR-22, miR-22-5p, or miR-22-3p mature sequences. Nevertheless, we can still conclude that miR-22 vs. miR-212 and miR-132 have opposite roles in breast cancer, with miR-22 showing oncogenic traits and the other two miRNAs oncosuppressive ones.

In support of a general proto-oncogenic role, it is worth mentioning that miR-22 has been found up-regulated in blood samples of all analyzed cancers according to dbDEMC 3.0: brain cancer (5 studies out of 5), breast cancer (3 studies out of 3), colorectal (3 studies out of 3) and gastric cancer (4 studies out of 5), hepatocellular carcinoma (4 studies out of 4), lung cancer (4 studies out of 5), and prostate cancer (3 studies out of 3).

## 6. Future Perspectives

Based on the extensive literature published so far, it is apparent that miR-22 has caught enormous attention in the cancer research field and beyond. This is fully justified by the powerful biological outcomes triggered by miR-22 overexpression or inactivation. However, it is also clear that additional rigorous studies in vivo in immunocompetent animal models are needed in order to have a more definitive view of its role in various tumor types.

In this respect, it must be underscored that miR-22 KO (knockout) mice have not been found to be tumor prone in several studies [90,91,92], as also reported in [93] where the authors have monitored miR-22 deficient mice for about 5 years. Instead, miR-22 KO (knockout) mice have been found to be tumor-resistant [34,39]. Conversely, in each immunocompetent transgenic model reported in the literature so far, miR-22 is consistently found to be oncogenic [34,39]. Critically, in the tumor types where miR-22 is regarded as tumor suppressive, studies performed in robust and immunocompetent model systems, or KO mice, have not been reported to date. This void must be filled in future efforts, ideally through the systematic use of immunocompetent and tissue-specific conditional transgenic or KO mice.

The notion that an immunocompetent model system is required to study miR-22 is not only supported by the extensive literature we reviewed here, but also by the fact that miR-22 has already been discovered to play an important role in the immune system [24]. However, up to now, the roles of miR-22 in the immune system and in cancer have not yet been fully integrated. Since we now know the fundamental role of the immune system in cancer initiation and progression, miR-22 functions in tumorigenesis should, in fact, be studied by systematically integrating its role in the proper tumor and in an immune-competent microenvironment. Indeed, a recent paper has already shown a possible molecular mechanism by which miR-22 affects immune-system response in a tumorigenic context [94]. miR-22 impairs the anti-tumor ability of Dendritic Cells (DCs) by targeting p38. DCs play an important role in anti-tumor immune response, and in turn, tumor cells negatively affect DCs differentiation, as well as their ability to activate the immune response by secretion of soluble factors [95,96]. miR-22, when expressed in DCs, impairs their tumor suppressive function, directly regulating p38 at the post-transcriptional level [94]. In turn, it negatively affects IL6 transcript levels, and the differentiation of DC driven by Th17 cells [94]. Interestingly, it has also been demonstrated that miR-22 represses the Th17 cell’s function by targeting PTEN-regulated pathways [97]. So, miR-22 acts as an oncomiR carrying out its action at two levels: directly in tumor cells by the mechanisms described above and by “antagonizing” the immune system response against cancer cells [94,97]. On the basis of this notion, secreted and circulating miR-22 might play a critical role in opposing an immunocompetent microenvironment.

Another consideration to keep in mind when trying to shed light on the role of miR-22 and other miRNAs in cancer is that scientific conclusions are mostly dependent on the number and the quality of the controls used, as well as the number of experiments performed using different techniques. For sure, the complexity of miR-22 is inherent to its miRNA nature since each miRNA can target multiple genes at the same time. Additionally, miRNA targets can themselves be oncogenes and tumor suppressors. For example, one of the strongest miR-22 targets is ER*α* [48], which is regarded both as an oncogene [39] and as a tumor suppressor [48] depending on the cellular and genetic context and the tumor evolution of breast cancer. The role of any miRNA is always a game of balance among all the actors, and this is particularly true for miR-22. For these reasons, in order to translate the biological and biochemical information about miR-22 into therapeutic approaches, it will be very important to do a zoom-out and consider the whole organismal dimension in which miR-22 is expressed or targeted for therapy. Indeed, the important oncogenic role of miR-22 in several tumor types is clearly unraveled in spontaneous and immunocompetent in vivo model systems, where this whole organismal dimension is preserved.

All these aspects at stake transcend a mere scholarly debate because miR-22, as other oncomiRs, is highly druggable using LNAs or other approaches. Thus, while further studies will obviously be needed to fully elucidate the role of miR-22 in cancer, preclinical trials in a model system and clinical trials in humans will tell if miR-22 therapeutic targeting can be effectively exploited for cancer treatment and the treatment and prevention of other diseases.

## Figures and Tables

**Figure 1 cancers-14-06255-f001:**
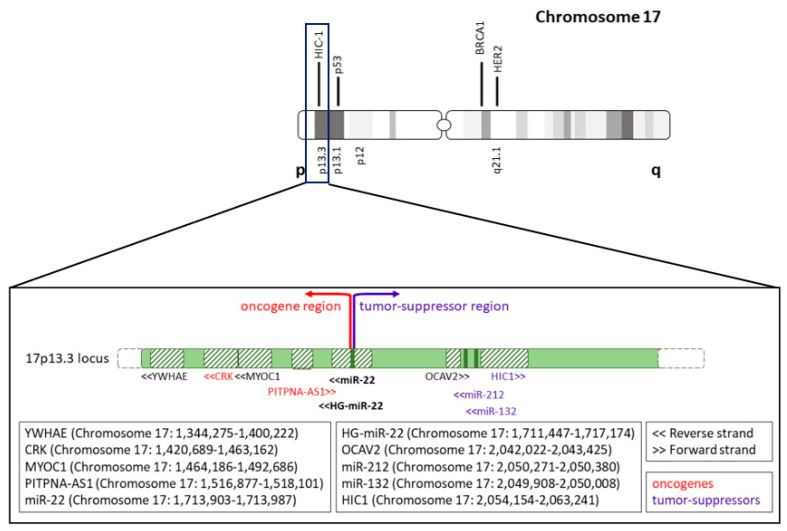
Schematic representation of miR-22 genomic locus on chromosome 17.

**Figure 2 cancers-14-06255-f002:**
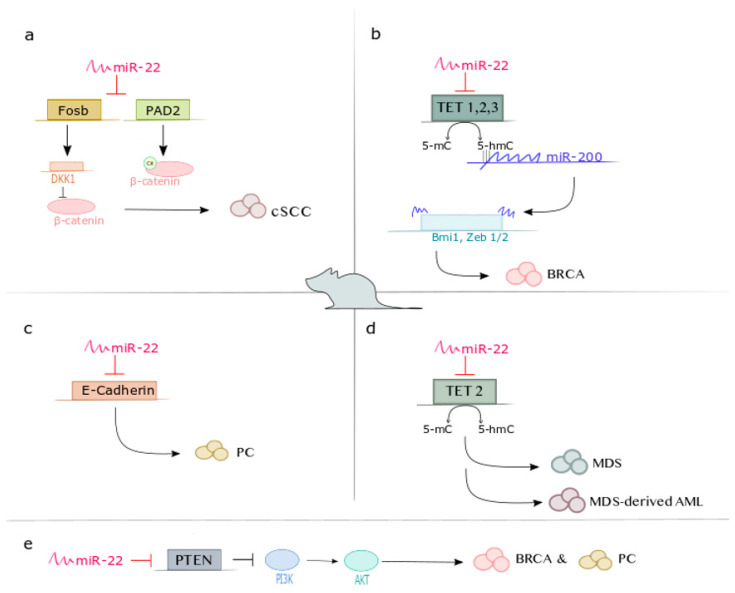
Mir-22 oncogenic role in cancer. Scheme of miR-22 functional mechanisms in promoting tumorigenesis of cutaneous squamous cell carcinoma (cSCC), breast cancer (BRCA), prostate cancer (PC), and Leukemia in immunocompetent (**a**–**d**) and xenograft (**e**) mouse models. (**a**) miR-22 promotes cSCC initiation and progression, repressing Wnt/β-catenin signaling by targeting Fosb and PAD2. (**b**) Epigenetic inactivation of miR-200 through miR-22 targeting the TET family in breast cancer triggers EMT and increases mammary tumorigenesis and metastasis. (**c**) miR-22 promotes tumor invasion in prostate cancer targeting E-Cadherin. (**d**) miR-22 decreases the level of 5-hmC by negatively regulating TET2 leading to MDS and MDS-derived leukemia. (**e**) Increased levels of miR-22 directly target PTEN, which results in PI3K/AKT signaling pathway upregulation and cancer progression in Breast and Prostate in vivo mouse models.

**Figure 3 cancers-14-06255-f003:**
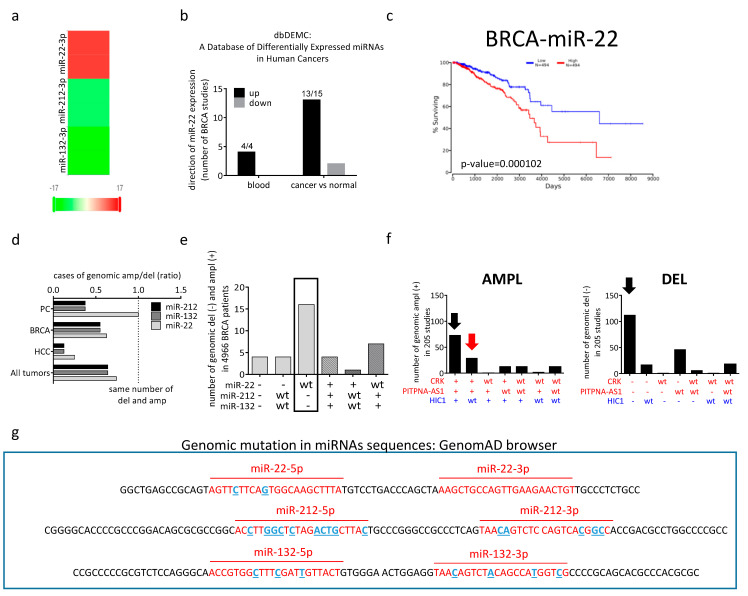
Alterations in miR-22 expression level, genomic location, and sequence. (**a**) Differential expression profile of miR-22-3p (miR-22), miR-132-3p (miR-132), and miR-212-3p (miR-212), in breast cancer, according to dbDEMC 3.0 database. (**b**) The number of breast cancer studies where miR-22 expression has been found up-regulated (black) or down-regulated (grey), according to dbDEMC. (**c**) Breast cancer samples available at TCGA (OncoLnc analysis) were divided at the median expression levels of hsa-miR-22-3p. The survival curve was then calculated for high (above the median, red) and low (below the median, blue) expression. (**d**) The ratio of the number of amplifications (+) and deletions (−) reported on cBioportal database for the indicated microRNAs in all tumors and in breast cancer (BRCA), hepatocellular carcinoma (HCC) and prostate cancer (PC). (**e**) The number of amplifications (+) and deletions (−) of miR-22, miR-132, and miR-212 in BRCA patients reported in the cBioportal database. (**f**) The number of amplifications (left) and deletions (right) reported on the cBioportal database for *CRK*, *PITPNA-AS1,* and *HIC1* in 205 studies. See the text for details about the cases highlighted by the black and red arrows. (**g**) Base pair variations in miRNA sequences reported in GenomAD browser. Red: mature miRNA sequence; blue: base pair variation; black: pre-miRNA sequence.

**Table 1 cancers-14-06255-t001:** Validated miR-22 target genes in cancer.

Target Gene	Tumor	Pathway	Reference
SIRT1	breast, ovarian, cervical, HCC	pRB	21502362 (Xu, D., 2011); 28882183 (Zhang, X., 2017)
CDK6	breast	pRB	21502362 (Xu, D., 2011)
CD147	breast	miR-22/Sp1/c-Myc	24906624 (Kong, L.M., 2014)
ERa	breast	PKC/ERK	19414598 (Pandey, D.P., 2009); 33173749 (Gao, Y., 2020)
Erbb3	lung	PI3K/AKT	22484852 (Ling, B., 2012)
WNT1	GC	Wnt/β-catenin	23851184 (Tang, H., 2013)
CD151	GC	-	24495805 (Wang, X., 2014)
CDKN1A	HCC	-	23582783 (Shi, C., 2013)
NCOA1	HCC	NF/kB	21798241 (Takara, A., 2011)
GLUT1	breast	-	25304371 (Chen, B., 2015)
SP1	breast, cervical, GC, HCC	pRB	21502362 (Xu, D., 2011); 23529765 (Guo, M.M., 2013).
HDAC4	HCC	-	20842113 (Zhang, J., 2010) 23349832 (Jovicic, A., 2013)
MYCBP	breast	MYCBP/c-Myc	20562918 (Xiong, J. 2010)
HDAC6	cervical	E6/p53	30379969 (Wongjampa, W. 2018)
LGALS1	RCC	HIF/mTOR	24496460 (White, N.M., 2014)
HIF1A	RCC	HIF/mTOR	24496460 (White, N.M., 2014)
NET1	CML	Net1/RhoA	25041463 (Ahmad, H.M., 2014)
TIAM1	CRC	Rac/Rho	23440286 (Li, B., 2013)
BTG1	CRC	BTG1/LC3II; apoptosis	25449431 (Zhang, H., 2015)
LGALS9	HCC	Tim3/Gal-9	26239725 (Yang, Q., 2015)
MMP14	GC	ECM, EMT	26610210 (Zuo, Q.F., 2015)
SNAIL	GC	ECM, EMT	26610210 (Zuo, Q.F., 2015)
ACLY	osteosarcoma, prostate, cervical, lung, breast	FASN/HMGCR	27317765 (Xin, M., 2016); 29636857 (Liu, H., 2018)
TIP60 *	breast	EMT	26512777 (Pandey, A.K., 2015)
NRAS *	breast	PI3K/AKT, MAPK/ERK, NF-kB	30384806 (Song, Y.K., 2018)
TWIST1 *	osteosarcoma	EMT	32391253 (Zhu, S.T., 2020)
TET2	MDS, breast, HCC	Epigenetic	23827711 (Song, S.J., 2013) 23830207 (Song, S.J., 2013) 34019487 (Chen, D., 2021)
PTEN	prostate, breast	PI3K/Akt	20388916 (Poliseno, L., 2010) 33173749 (Gao, Y., 2020)
CBL *	papillary thyroid	Wnt/β-catenin	30190130 (Wang, M., 2018)
FOSB *	cSCC	Wnt/β-catenin	34345013 (Yuan, S., 2021)
PAD2 *	cSCC	Wnt/β-catenin	34345013 (Yuan, S., 2021)

All reported targets are strongly validated by reporter assay and some even by western blot or qPCR. All of them are present in “https://mirtarbase.cuhk.edu.cn/ (accessed on 15 November 2022)”, except for those with the (*) that have been collected from the literature. Oncogenic and tumor-suppressive targets are reported in red and blue, respectively. GC (gastric cancer), HCC (hepatocellular carcinoma), RCC (renal cell carcinoma), CML (chronic myeloid leukemia), CRC (colorectal cancer), MDS (myelodysplastic syndrome), cSCC (cutaneous squamous cell carcinoma).

**Table 2 cancers-14-06255-t002:** miR-22 role in several *in vivo*-based cancer studies.

Tumor	miR-22 Role	Xenograft	Trangenic	Knock-Out	Reference
AML	tumor suppressor	✔			Shen, C. et al., 2016
	tumor suppressor	✔		✔ *	Jiang, X. et al., 2016
Breast Cancer	oncogene		✔	✔ *	Gao et al., 2020
	oncogene	✔	✔		Song et al., 2013
	tumor suppressor	✔			Gorur, A. et al., 2021
	tumor suppressor	✔			Liu, X. et al., 2018
	tumor suppressor	✔			Liu, H. et al., 2018
	tumor suppressor	✔			Shao, P. et al., 2017
	tumor suppressor	✔			Kong, L.M. et al., 2014
	tumor suppressor	✔			Xu, D. et al., 2011
Colon Cancer	tumor suppressor	✔			Cong, J. et al., 2020
	tumor suppressor	✔			Sun, R. et al., 2019
	tumor suppressor	✔			Hu, Y., 2019
	tumor suppressor	✔			Liu Y., 2018
	tumor suppressor	✔			Xia, S.S. et al., 2017
	tumor suppressor	✔			Zhang, H. et al., 2015
cSCC	oncogene	✔		✔ *	Yuan, S. et al., 2021
gastric cancer	tumor suppressor	✔			Zong, W. et al., 2020
	tumor suppressor	✔			Li, X. et al., 2020
	tumor suppressor	✔			Gan, L. et al., 2019
	tumor suppressor	✔			Li, S. et al., 2018
	tumor suppressor	✔			Tang, H. et al., 2013
Hepatocellular Carcinoma	oncogene	✔			Chen, D. et al., 2021
	tumor-suppressor	✔			Zhang, L. et al., 2021
	tumor suppressor	✔			Chen, F., 2020
	tumor suppressor	✔			Chen, S. et al., 2020
	tumor suppressor	✔			Leung, Z. et al., 2019
	tumor-suppressor	✔			Zhao, L. et al., 2019
	tumor suppressor	✔			Chen, S. et al., 2018
	tumor-suppressor	✔			Yang, F. et al., 2016
	tumor-suppressor	✔			Zhang, J. et al., 2010
MDS	oncogene		✔		Song, S.J. et al., 2013
Osteosarcoma	tumor suppressor	✔			Xue, Y. et al., 2021
	tumor-suppressor	✔			Meng, C.Y. et al., 2020
	tumor-suppressor	✔			Meng, C.Y. et al., 2020
	tumor-suppressor	✔			Zhu, H. et al., 2020
Prostate Cancer	oncogene		✔		Hemani, R. et al., 2022
	oncogene		✔		Joshi, T. et al., 2020
	oncogene		✔		Dhar, S. et al., 2017
	oncogene	✔			Budd, W.T. et al., 2015
	oncogene		✔		Poliseno, L. et al., 2010

* Immunodeficient or * Immunocompetent mouse models for miR-22 knock-out study. MDS (myelodysplastic syndrome); AML (acute myeloid leukemia); cSCC (cutaneous squamous cell carcinoma).

**Table 3 cancers-14-06255-t003:** The number of amplifications and deletions reported in the cBioportal database (cbioportal.org) for the three microRNAs (miR-22, miR-132, and miR-212) in all tumors and, in breast cancer (BRCA), in hepatocellular carcinoma (HCC), and in prostate cancer (PC).

Tumor Type	Amplification/Deletion	miR-22	miR-132	miR-212
**All tumors**	amp	97	94	94
	del	131	146	146
	ratio	0.74	0.64	0.64
**HCC**	amp	2	1	1
	del	8	8	8
	ratio	0.25	0.125	0.125
**BRCA**	amp	5	11	11
	del	8	20	20
	ratio	0.625	0.55	0.55
**PC**	amp	8	3	3
	del	8	7	7
	ratio	1	0.375	0.375

**Table 4 cancers-14-06255-t004:** Occurrence in genetic alterations of miR-22, miR-212, miR132, HIC1, and miR22HG genes reported in cbioportal.org.

	A	B	Neither	A Not B	B Not A	Both	*p*-Value	Tendency	Fisher Test
**1**	miR-22	miR-132	41760	36	36	177	<0.001	Co-occurrence	*p*-value > 0.05		*p*-value
**2**	miR-22	miR-212	41762	36	34	177	<0.001	Co-occurrence	*p*-value < 0.05	
**3**	miR-132	miR-212	41796	2	0	211	<0.001	Co-occurrence		<0.05
**4**	miR22HG	miR-22	41769	27	8	205	<0.001	Co-occurrence			
**5**	miR22HG	miR-132	41761	35	16	197	<0.001	Co-occurrence			
**6**	miR22HG	miR-212	41763	35	14	197	<0.001	Co-occurrence			
**7**	HIC1	miR-22	47611	73	38	190	<0.001	Co-occurrence			
**8**	HIC1	miR-132	47648	23	1	240	<0.001	Co-occurrence			
**9**	HIC1	miR-212	47648	25	1	238	<0.001	Co-occurrence

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
