# Peer review of "An Immunocompetent Environment Unravels the Proto-Oncogenic Role of miR-22"

_cancers, 2022, doi:10.3390/cancers14246255_

Round 1
Reviewer 1 Report
miR-22 has been extensively studied because of its powerful and diverse biological activities in various cancers. But it is still missing about the controversial role of miR-22 that it can act as both an oncogene and a tumor suppressor in the same cancer. In this paper, the authors shed light on this scientific mystery. The authors highlight that an immunocompetent mouse model was critical to obtaining more reliable research results. A lot of results based on immunodeficient mouse model are utterly confusing and even contradictory. In addition, the authors focus on three different tumors as paradigmatic examples (breast cancer, prostate cancer, and leukemia).
I am appreciation with the authors’ opinion. Tumor is composed of tumor cells and tumor immune microenvironment cells and is developed under the supervision of immune cells. And most of the genes are adaptive expression genes and their function can be affected by environment. So miR-22’s function is theoretically different between immunocompetent mouse model and immunodeficient mouse model. But the real environment in which miR-22 functions is under normal immune conditions. At the same time, the authors seem to give a warning to researchers that it is best to select immunocompetent mouse model to get reliable results in the study of gene function in cancer. This is also the most important significance of this paper.
However, whether the controversial role of miR-22 is attribute to the difference in tumor immune microenvironment? The direct evidence is lacking. The authors should supply more relative evidence or references to support or enrich their point of view.
Another point confused me is that I do not know the necessity of the part about analysis of miR-22’s genomic locus. How this part support the topic “An immunocompetent environment unravels the proto-oncogenic role of miR-22”?
Author Response
Reviewer 1
miR-22 has been extensively studied because of its powerful and diverse biological activities in various
cancers. But it is still missing about the controversial role of miR-22 that it can act as both an oncogene
and a tumor suppressor in the same cancer. In this paper, the authors shed light on this scientific mystery.
The authors highlight that an immunocompetent mouse model was critical to obtaining more reliable
research results. A lot of results based on immunodeficient mouse model are utterly confusing and even
contradictory. In addition, the authors focus on three different tumors as paradigmatic examples (breast
cancer, prostate cancer, and leukemia).
I am appreciation with the authors’ opinion. Tumor is composed of tumor cells and tumor immune
microenvironment cells and is developed under the supervision of immune cells. And most of the genes
are adaptive expression genes and their function can be affected by environment. So miR-22’s function is
theoretically different between immunocompetent mouse model and immunodeficient mouse model. But
the real environment in which miR-22 functions is under normal immune conditions. At the same time, the
authors seem to give a warning to researchers that it is best to select immunocompetent mouse model to
get reliable results in the study of gene function in cancer. This is also the most important significance
of this paper.
• However, whether the controversial role of miR-22 is attribute to the difference in tumor immune
microenvironment? The direct evidence is lacking. The authors should supply more relative evidence or
references to support or enrich their point of view.
We thank the reviewer for prompting us to supply more relative references about the direct link between
the immune system, miR-22 and cancer cells. As we now discuss in the revised review, miR-22 plays an
important role in the immune system and in cancer cells. However, up to now, the roles of miR-22 in the
immune-system and in cancer have not yet been fully integrated. As we now have described in the paragraph
“Future perspectives”, a recent paper shows a possible molecular mechanism by which miR-22 affects
immune-system response in a tumorigenic context (Liang et al., 2015). Particularly, miR-22 impairs the antitumor ability of Dendritic Cells by targeting p38. This is just one of the molecular mechanisms by which miR22, carrying out its role in a tumorigenic context, affects immune system response. Probably, many others
will be identified in the future.
• Another point confused me is that I do not know the necessity of the part about analysis of miR-22’s
genomic locus. How this part support the topic “An immunocompetent environment unravels the protooncogenic role of miR-22”?
We thank the reviewer for this comment. The miR-22 genome locus/gene expression analysis described in
the paragraph “Alterations of miR-22 gene” has helped us to better understand its role in tumorigenesis.
Particularly, we have compared the miR-22 gene with cancer relevant genes that are telomeric (CRK and
PITPNA-AS1) or centromeric (miR-132, miR-212, and HIC1) to it. It is well known that CRK and PITPNA-AS1
genes act as oncogenes, while miR-132, miR-212, and HIC1 act as tumor-suppressor genes. Therefore,
through this analysis, we have tested if miR-22 is co-selected more closely with the oncogenic/telomeric
genes or with the tumor-suppressor/centromeric genes. All the data collected, although some of them are
only tendencies in view of the low number of samples, lend support to the oncogenic role of miR-22.
Therefore, this paragraph is important to support the oncogenic role of miR-22, while the importance of the
immune system for its study, is brought about in the following paragraphs. To highlight this point, we have
now added a phrase at the beginning of the paragraph.
Reviewer 2 Report
In this review, Maria Laura Centomo et al comprehensively reviewed the diverse biological activities in different cancers focusing on the contrasting results concerning its ongenic vs tumor-suppressive nature. While the majority of evidence presented is convincing, comparing miR-22 amp/deletion ratio with that of miR-132 or miR-212 to support that miR-22 is an oncogene is rather superficial. However, the overall quality of this review article is excellent based on the analyses of its genomic locations and alterations and the extensive literature review. There are a few gramma errors that need to be fixed. Finally, some of the figure legends are too small to read.
Author Response
In this review, Maria Laura Centomo et al comprehensively reviewed the diverse biological activities in
different cancers focusing on the contrasting results concerning its ongenic vs tumor-suppressive nature.
While the majority of evidence presented is convincing, comparing miR-22 amp/deletion ratio with that of
miR-132 or miR-212 to support that miR-22 is an oncogene is rather superficial. However, the overall
quality of this review article is excellent based on the analyses of its genomic locations and alterations and
the extensive literature review. There are a few gramma errors that need to be fixed.
We thank the reviewer for this comment. As for the miR-22 genome locus/gene expression analysis described
in the paragraph “Alterations of miR-22 gene”, it has helped usto better understand its role in tumorigenesis.
Particularly, we have compared the miR-22 gene with cancer relevant genes that are telomeric (CRK and
PITPNA-AS1) or centromeric (miR-132, miR-212, and HIC1) to it. It is well known that CRK and PITPNA-AS1
genes act as oncogenes, while miR-132, miR-212, and HIC1 act as tumor-suppressor genes. Therefore,
through this analysis, we have tested if miR-22 is co-selected more closely with the oncogenic/telomeric
genes or with the tumor suppressor/centromeric genes. All the data collected, although some of them are
only tendencies in view of the low number of samples, lend support to the oncogenic role of miR-22.
Finally, some of the figure legends are too small to read.
Thank you for pointing this out. We have now increased the size of the figure legends.